# Effects of Chlorothalonil Application on the Physio-Biochemical Properties and Microbial Community of a Yellow–Brown Loam Soil

Jinlin Jiang *, Yuwen Yang, Lei Wang, Shaohua Cao, Tao Long and Renbin Liu

Key Laboratory of Soil Environmental Management, Nanjing Institute of Environmental Sciences, Ministry of Ecology and Environment of the People's Republic of China, Nanjing 210042, China; yyw@nies.org (Y.Y.); wanglei@nies.org (L.W.); csh@nies.org (S.C.); longtao@nies.org (T.L.); lrb@nies.org (R.L.)
* Correspondence: jjl@nies.org

**Abstract:** To gain better knowledge of the effects of residual chlorothalonil on soil characteristics and soil microbial communities, we evaluated the dissipation of chlorothalonil and the effects of different chlorothalonil concentrations on soil respiration, enzyme activities, and microbial community structure in yellow–brown loam soils. Bacterial and fungal soil communities were examined using traditional plate counting and polymerase chain reaction–denaturing gradient gel electrophoresis (PCR–DGGE) methods. Soil properties and the results of DGGE band analysis were both used to estimate the status of the soil microbial ecosystem. The results show that residual chlorothalonil has considerable effects on soil respiration, enzymatic activities, and microbial community structure. In particular, soil respiration and phosphatase activities were increased, while saccharase activity, microbial biomass, and microbial community diversity were decreased by increasing levels of chlorothalonil treatment. Correlation analyses revealed that the application of chlorothalonil was significantly correlated with the change of the soil respiration, urease activity, sucrase activity, soil culturable bacteria, and culturable fungi biomass. We conclude that residual chlorothalonil is directly related to soil respiration, enzyme activities, and microbial community structure.

**Keywords:** fungicide; dissipation; soil respiration; soil enzyme; soil microbial population

## 1. Introduction

The use of fungicides, especially chlorothalonil, is a critical tool for the management of fungal diseases in agricultural crops, recreational turf, and urban lawns [1]. Each year, an excess of $4.8 \times 10^6$ kg of chlorothalonil is applied to field crops such as peanuts, vegetables, and fruits, according to the estimation of United States Geological Survey (USGS) [2]. Chlorothalonil was introduced by Diamond Shamrock Corp in 1965 and was firstly registered in 1966. This chemical is a broad-spectrum fungicide with characteristics of low aqueous solubility and high soil sorption capacity [3]. It is usually applied 1−3 times to the same crop within a growing season, with an application interval of less than 10 days. Studies on its dissipation behavior in and its effects on soil have revealed that repeated applications of high concentrations of chlorothalonil in soil results in an accumulation after crop harvesting [4–6]. Because chlorothalonil is not broken down by ultraviolet light, its residual activity persists for a relatively long time. Moreover, because chlorothalonil is highly toxic to fish and aquatic invertebrates, its transport and toxicity are matters of great concern, especially in aquatic systems [7].

Yellow–brown loam soil is widely distributed in China, with superior hydrothermal conditions and high natural fertility. Therefore, the yellow–brown loam soil area is an intense production area of economic forest in China and suitable for the growth of a variety of trees. It is also an important agricultural area, rich in a variety of food and cash crops.

---

However, characteristics of the effects of residual chlorothalonil on essential soil micro-ecosystem functions and properties, including soil respiration, soil enzymatic activities, microbial biomass, and microbial community structure, as well as the involved mechanism remain obscure. Pesticides are known to interact with microorganisms in ways that can both negatively or positively affect the soil microbiota [8]. Investigating the connections between critical soil processes and microbial community composition can also facilitate our understanding of the roles played by the soil biota, biodiversity, and soil microbial community components in safeguarding soil ecosystems from residual fungicides. Soil respiration and enzymatic activities can also reflect the biomass and structure of microbial communities and are considered integrative bio-indicators [9]. They are therefore often used to monitor the impacts of soil management (e.g., including agricultural practices, pesticide application, and other contaminants) on soil health [10,11]. Enzymes are sensitive indicators of soil quality. However, their activities vary naturally, and this needs to be considered when interpreting their activity levels, especially in contaminated soils [12]. In this context, the changes in respiration, enzymatic activities (urease, phosphatase, and saccharase), microbial biomass, and microbial community structure of yellow–brown earth soils in response to residual chlorothalonil were investigated to accumulate basic data for studying the influence of residual pesticides on the structure, function, and processes of soil ecosystem.

## 2. Material and Methods

### 2.1. Soils

The experiment was conducted on soil collected from the top layer (0–15 cm) of a botanical garden located in Nanjing, China (31°14′ N, 118°22′ E). The soil was identified as yellow–brown loam soil, which contained no detectable amounts of residual chlorothalonil, with the following physicochemical characteristics: total nitrogen 757.4 mg·kg$^{-1}$, total carbon 12.7 g·kg$^{-1}$, available phosphorus 334.6 mg·kg$^{-1}$, available potassium 116.3 mg·kg$^{-1}$, moisture content 28.5%, and pH 6.7.

### 2.2. Chemicals

Chlorothalonil (99.5% purity) was purchased from Sigma (Sigma-Aldrich Inc., St. Louis, MO, USA). All other reagents were of analytical grade. Chromatographically pure n-hexane, ethyl acetate, n-hexane-dichloromethane, and acetone were purchased from Merck (Darmstadt, Germany).

### 2.3. Soil Treatment and Sampling

Chlorothalonil was added to soil to obtain final nominal concentrations of 3.0 and 300 mg·kg$^{-1}$ of dry soil (treatments Ch 3 and Ch 300, respectively) corresponding to the recommended (3000 g of active ingredient per ha) and 100 times the recommended dose for agricultural use, respectively. To distribute this chemical evenly, the soil samples were thoroughly mixed with a sterile steel spoon and passed through a 2 mm sieve. Soil without chlorothalonil acted as the control treatment (Ck). The treated soils (2.0 kg) were placed in 3-L polypropylene pots and incubated for 60 days at 25.6 ± 1.8 °C in the dark. Samples from the control and treatment groups were collected after 0, 10, 20, 30, and 60 days to determine the residual levels of chlorothalonil in soil, soil respiration, and enzyme activities (urease, phosphate, and invertase). We also evaluated the changes in the soil microbial community represented by the culturable microbial biomass and polymerase chain reaction–denaturing gradient gel electrophoresis (PCR–DGGE) bands via the plate counting method and PCR–DGGE analysis, respectively. All procedures were carried out in triplicate.

### 2.4. Measurement of Chlorothalonil in Soil

Ten grams of soil were ultrasonicated in an ultrasonic bath with 40.0 mL of n-hexane-dichloromethane (1:1, $v/v$) for 20 min. The mixture was vacuum-filtered with a filter paper (0.45 μm) on a Buchner funnel. The filtrate was re-extracted with 40.0 mL of n-hexane-

dichloromethane (1:1, $v/v$), and the filter cake was twice washed with 10.0 mL of n-hexane. The pooled filtrates were concentrated to roughly 1 mL using a rotary evaporator after being passed through anhydrous sodium sulfate. Prior to gas chromatographic analysis, the residual was further concentrated to near dryness under a mild nitrogen flow and lastly dissolved in 10.0 mL n-hexane. According to Wu et al. [4], gas chromatography was used to measure chlorothalonil using an Agilent 7890A gas chromatograph coupled with an electron capture detector (GC–ECD).

The average recoveries of chlorothalonil on spiked (0.3, 3, 30, and 300 mg·kg$^{-1}$ of dry soil) soil varied from 80.46% to 92.42%, while the relative standard deviation was less than 3.62%. The detection and quantification limits of the method were determined to be 0.001 mg·kg$^{-1}$ and 0.01 mg·kg$^{-1}$ of dry soil, respectively.

### 2.5. Determination of Soil Respiration and Enzymatic Activities

Soil respiration was measured using the modified substrate-induced respiration method according to Höper [13]. Glucose was added to 1 g soil in a glass vial (10 mL) at a concentration of 10 mg·g$^{-1}$ soil and well mixed. The vial was sealed and incubated at 25 °C for 1 h. Soil respiration was measured in terms of carbon dioxide ($CO_2$) evolution, which was determined by comparing the initial and final $CO_2$ concentrations measured using a soil respiration meter (China) [14].

Soil urease activity was determined spectrophotometrically using urea as a substrate [15]. First, 1.0 g soil was placed in a 50 mL bottle, and 1 mL toluene, 10 mL 10% urea solution, and 10 mL 0.1 M $Na_2HPO_4$-citric acid buffer (pH 6.7) were added and kept in a water bath at 37 °C. After 24 h incubation, 4 mL sodium phenate and 3 mL sodium hypochlorite were added, and then the solution was diluted to 20 mL with deionization water. $NH_4^+$-N concentration was determined by the spectrophotometer at a wavelength of 578 nm. A series of standard solutions of $NH_4Cl$ covering the concentrations of 0.05–2.0 mg $NH_4^+$-N per mL was prepared for calibration. Urease activity was expressed as the amount of $NH_4^+$-N released from urea per gram dried soil after incubation at 37 °C for 24 h.

Soil phosphatase activity was determined using disodium phenyl phosphate as a substrate [16]. A total of 1.0 g of soil was placed in a 50 mL bottle, and 0.2 mL of toluene, 5 mL of 0.5% disodium phenyl phosphate solution, and 5 mL of 0.1 M phosphate buffer (pH 7.0) were added to the flask, and the suspensions were kept in a water bath at 37 °C for 4 h. After incubation, suspensions of 1 mL were moved to a tube (15 mL), and 2.5 mL of 0.5% 4-aminoantipyrine, 2.5 mL of 2.5% red potassium prussiate, and 10 mL of deionization water were added. The flask was then swirled for a few seconds to mix the contents. After 30 min, the filtrate was determined by a spectrophotometer at a wavelength of 570 nm. A series of standard solutions of phenol covering the concentrations of 0.05–5.0 mg phenol per mL was prepared for calibration. Phosphatase activity was expressed as the amount of phenol released per gram dried soil after incubation at 37 °C for 24 h.

Soil saccharase activity was determined using 3,5-dinitrosalicylic acid as a substrate [17]. A total of 2.0 g of soil sample was placed in a 50 mL bottle and mixed with 15 mL of 18% sucrose solution, 5 mL of phosphate buffer (pH 5.5), and 0.2 mL of methylbenzene, and then kept in a water bath at 37 °C for 24 h. Then, 3,5-dinitrosalicylic acid solution and deionized water were added into the filtrate in the tubes, and the color intensity was determined using a spectrophotometer at a wavelength of 508 nm. Simultaneously, controls with no-matrix and no-soil were made. Soil saccharase activity was expressed as the amount of glucose released per gram dried soil after 24 h of incubation at 37 °C.

### 2.6. Plate Counting and PCR–DGGE Analysis of the Soil Microbial Population

The total number of culturable microorganism in samples was determined by traditional dilution plate counting. A total of 100 μL of the diluted soil suspension was inoculated into an agar plate containing suitable medium. Bacteria and fungi were cultured on beef extract–peptone medium (BPM: 3 g beef extract, 10 g peptone, 15 g agar, and 10 g NaCl in 1 L distilled water) and potato dextrose agar medium (PDA: 11 g potato infusion,

20 g dextrose, and 15 g agar in 1 L distilled water), respectively. Three plates were prepared for each soil sample concentration, and the plates were incubated for 3 d at 28 °C. Plates were checked every day. The total number of microorganisms was determined by counting the number of colony forming units. The number of colonies per gram of dry soil in the original soil sample was calculated by multiplying the average colony count per dish by the dilution factor.

Soil DNA was isolated from 0.5 g of soil samples according to the manufacturer's instructions using an UltraClean Soil DNA Isolation Kit (MoBio, Carlsbad, CA, USA). The universal forward primer was used to amplify the bacterial 16S and fungal 18S rRNA genes (Table 1). PCR and DGGE were carried out according to the method described by Pan et al. [18]. The fingerprints obtained from DGGE were digitized, analyzed, and compared using Gel Compare II software (Applied Maths, Sint-Matenslatem, Belgium). Cluster analysis of the DGGE fingerprints expressed with the dendrograms was based on the similarity coefficient results, and soil microbial diversity analysis was based on the Shannon index results, which were both calculated using the Gel Compare II programs. The similarity coefficient was obtained from the following equation: $S_{ab} = 2N_{ab}/(N_a + N_b)$, where $S_{ab}$ is the similarity coefficient between soil DNA samples a and b; $N_{ab}$ is the numbers of shared co-migrating DGGE fragments; and $N_a$ and $N_b$ are the total numbers of DGGE fragments from soil DNA samples a and b, respectively.

**Table 1.** PCR primers used in the present study [19,20].

| Target Group | Primer Name [a] | Sequence (5′-3′) |
|---|---|---|
| Bacterial16S rRNA | GC-338F<br>518R | (GC) [b] -GACTCCTACGGGAGGCAGCAG<br>ATTACCGCGGCTGCTGG |
| Fungal 18S rRNA | CG-Fungi F<br>Fungi R | (GC) [c] -ATTCCCCGTTACCCGTTG<br>GTAGTC ATATGC TTGTCTC |

[a], F represents forward primer. R represents reverse primer. [b], (GC): CGC CCG CCG CGC GCG GCG GGC GGG GCG GGG GCA CGG GGG. [c], (GC): CGCCC GCCGC GCCCC GCGCC GGCCC GCCGCC CCCGC CCC.

### 2.7. Statistical Analysis

Data were checked for deviations from normality and homogeneity of variance before analysis. A one-way ANOVA test was used to analyze the statistical significance with the software package of SPSS 17.0 (SPSS Inc., Chicago, IL, USA). Statistically significant differences corresponded to *p* values < 0.05, unless otherwise stated. The correlation analysis between chlorothalonil application and different treatments (soil respiration, enzymatic activities, and culturable microorganisms) were determined using general linear model analysis with SPSS 17.0. Model fitting of dissipation of chlorothalonil was conducted with OriginPro 2019 software. Data were presented as mean ± standard deviation.

### 3. Results

### 3.1. Dissipation of Chlorothalonil in Yellow–Brown Loam Soil

The dissipation of chlorothalonil in yellow–brown loam soil is shown in Figure 1. Approximately 98.8% and 96.2% of the initial chlorothalonil concentrations dissipated within 30 days in Ch 3 and Ch 300, respectively, with corresponding half-lives ($DT_{50}$) of 4.82 and 5.66 days, respectively. Chlorothalonil did not accumulate significantly in soil after single applications of 3.0 or 300 mg·kg$^{-1}$ of dry soil.

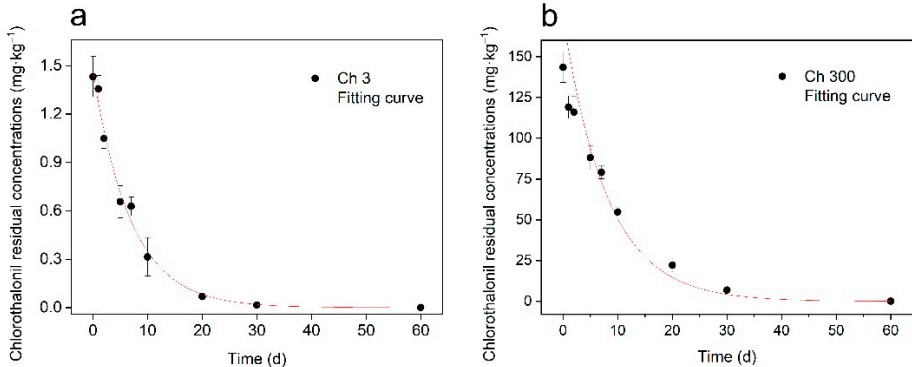

**Figure 1.** Dissipation of chlorothalonil in yellow–brown loam soil. (**a**) dissipation of chlorothalonil in soil after single applications of 3.0 mg·kg$^{-1}$ of dry soil; (**b**) dissipation of chlorothalonil in soil after single applications of 300 mg·kg$^{-1}$ of dry soil. All values represent mean ± standard deviation of triplicate samples. Ch 3 represents soil with 3.0 mg·kg$^{-1}$ chlorothalonil. Ch 300 represents soil with 300 mg·kg$^{-1}$ chlorothalonil. The same below.

### 3.2. Soil Respiration and Enzymatic Activities

Figure 2 shows the changes in the soil respiration and enzymatic activities during 60 days of incubation. At day 10, the levels of soil respiration in the Ck and Ch 3 treatments were significantly lower than those of Ch 300. Between 10 and 30 days, levels of soil respiration in Ch 3 and Ch 300 were significantly higher than those of Ck. Over the entire 60-day period, the levels of soil respiration in Ch 300 were higher than those in Ch 3 and Ck (Figure 2a).

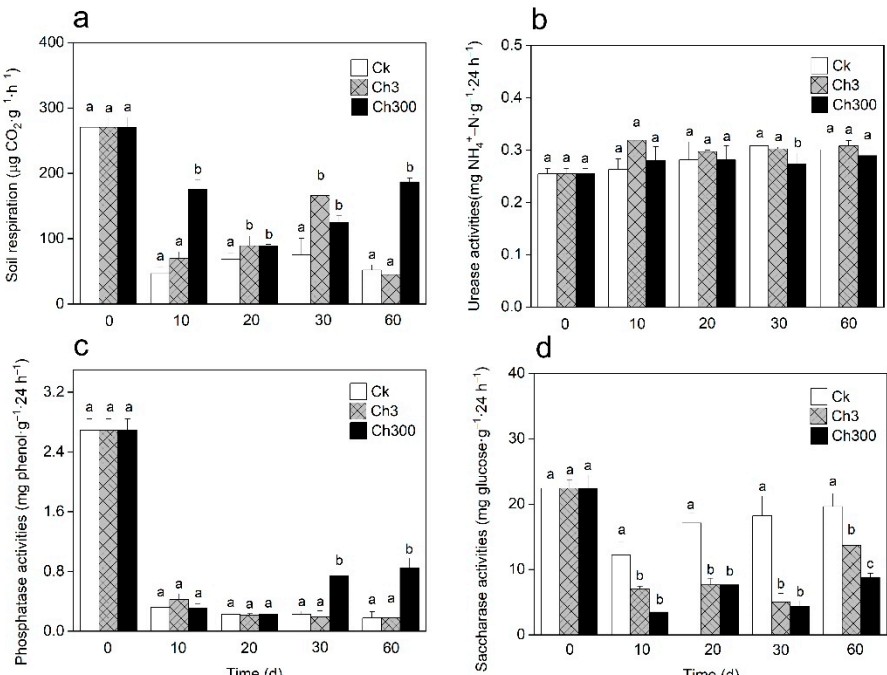

**Figure 2.** Changes in soil respiration and enzymatic activities of different treatments: (**a**) soil respiration; (**b**) urease activity; (**c**) phosphatase activity; (**d**) saccharase activity. Ck represents soil without chlorothalonil. Ch 3 represents soil with 3.0 mg·kg$^{-1}$ chlorothalonil. Ch 300 represents soil with 300 mg·kg$^{-1}$ chlorothalonil. All values represent mean ± standard deviation of triplicate samples.

Figure 2b shows similar changes of urease activities in all three treatments, except at day 30, when urease activity in Ch 300 was lower than those in Ch 3 and the control ($p < 0.05$). Similarly, as shown in Figure 2c, changes of phosphatase activities were similar in all three treatments, except that the levels in Ch 300 at day 30 and 60 were higher than those in Ch 3 and the control ($p < 0.05$). Saccharase activities in Ch3 and Ch 300 were significantly lower than those of the control in all treatments ($p < 0.05$), and at day 60, saccharase activity in Ch 300 was significantly lower than that of Ch 3 (Figure 2d).

### 3.3. Culturable Microbial Biomass

As shown in Table 2, the level of culturable bacterial biomass of the soils with all chlorothalonil treatment was significantly lower than that of the control at day 10 and 20 after application. After 30 days of treatment, the culturable bacterial biomass in Ch 3 had recovered, while that in Ch 300 was still significantly lower than Ck ($p < 0.05$). After 60 days, the culturable bacterial in Ch 300 had recovered.

The level of culturable fungal biomass in Ch 300 was significantly lower than that of the control and Ch 3 after 10 days ($p < 0.05$). After 30 days of treatment, the levels of fungal biomass in Ch3 and Ch 300 were all significantly lower than that of the control, and this phenomenon was still observed at day 60 after application.

**Table 2.** Abundance of culturable microorganisms in soil treated with chlorothalonil.

| Treatment | Bacteria ($10^5$) | | | | | Fungi ($10^3$) | | | | |
|---|---|---|---|---|---|---|---|---|---|---|
| | 0 d | 10 d | 20 d | 30 d | 60 d | 0 d | 10 d | 20 d | 30 d | 60 d |
| Ck | 7.75 ± 0.9 Aa | 8.25 ± 0.50 Aa | 5.67 ± 0.52 Aa | 8.33 ± 1.13 Aa | 7.83 ± 0.80 Aa | 1.15 ± 0.13 Aa | 1.17 ± 0.10 Aa | 0.90 ± 0.50 Aa | 3.08 ± 0.45 Ba | 2.93 ± 0.27 Ba |
| Ch3 | 7.75 ± 0.90 Aa | 5.5 ± 1.15 Bb | 4.42 ± 0.63 Bb | 7.5 ± 0.50 Aa | 5.75 ± 0.50 Aa | 1.15 ± 0.13 Aa | 1.39 ± 0.34 Aa | 1.08 ± 0.14 Aa | 1.39 ± 0.04 Ab | 1.89 ± 0.58 Ab |
| Ch300 | 7.75 ± 0.90 Aa | 1.78 ± 0.09 Bc | 2.33 ± 1.47 Ab | 2.12 ± 0.14 Bb | 5.33 ± 0.38 Aa | 1.15 ± 0.13 Aa | 0.63 ± 0.28 Ab | 0.83 ± 0.12 Aa | 1.83 ± 0.30 Bb | 1.04 ± 0.35 Ab |

Note: different lowercase letters in the same column indicate significant differences at $p < 0.05$; different uppercase letters in the same row indicate significant differences at $p < 0.05$. All values represent mean ± standard deviation of triplicate samples.

### 3.4. PCR–DGGE Profiles

To determine the effect of chlorothalonil on the microbial community structure, we monitored both bacterial and fungal communities through PCR–DGGE analysis (Figure 3 and Table 3). The results indicate that the bacterial communities changed in soil in response to the chlorothalonil treatment (Table 3 and Figure 3). As shown in Figure 3a, the bacterial communities in Ch 3 and Ch 300 changed obviously compared with those in the control at day 30, with similarities of less than 80% and 25%, which were less than 85% and 65% at day 10, respectively. Further banding pattern similarities analysis of PCR–DGGE profiles (Table 3) showed the Shannon index increasing from 2.69 to 2.94 and 2.20 to 3.01 at day 60 in Ch 3 and Ch 300, respectively, when compared to that at day 30. The fungal communities in Ch 3 and Ch 300 changed obviously compared with those in the control after 20 days, with similarity values of less than 20% and 25%, respectively (Figure 3b). The total species number (number of DGGE bands calculated by Gel Compare II software) and the Shannon index showed the same trend after 20 days, at which point the total fungal species number and the Shannon index in Ch 3 and Ch 300 were 16 and 2.14, and 7 and 1.36, respectively, while that in Ck were 17 and 2.23.

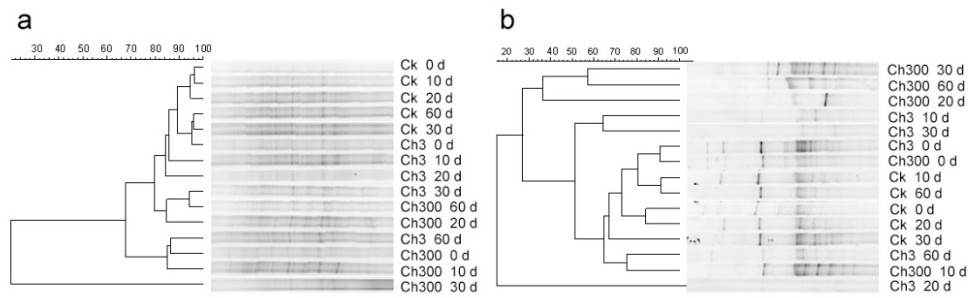

**Figure 3.** DGGE profile and analysis of soil bacterial and fungal communities: (**a**) Cluster analysis of soil bacteria from the DGGE pattern at different times. (**b**) Cluster analysis of soil fungi from the DGGE pattern at different times.

**Table 3.** Analysis and comparison of the DGGE fingerprints.

| Types | Index | Ck 0 d | Ck 10 d | Ck 20 d | Ck 30 d | Ck 60 d | Ch3 0 d | Ch3 10 d | Ch3 20 d | Ch3 30 d | Ch3 60 d | Ch300 0 d | Ch300 10 d | Ch300 20 d | Ch300 30 d | Ch300 60 d |
|---|---|---|---|---|---|---|---|---|---|---|---|---|---|---|---|---|
| Bacteria | Total species | 28 | 29 | 28 | 29 | 24 | 26 | 24 | 32 | 17 | 22 | 29 | 26 | 32 | 12 | 23 |
|  | Shannon | 2.90 | 3.07 | 3.23 | 3.15 | 2.95 | 3.05 | 3.09 | 3.14 | 2.69 | 2.94 | 3.19 | 3.09 | 3.31 | 2.20 | 3.01 |
| Fungi | Total species | 14 | 15 | 17 | 26 | 13 | 21 | 15 | 16 | 17 | 23 | 18 | 15 | 7 | 24 | 17 |
|  | Shannon | 2.15 | 2.22 | 2.23 | 2.82 | 1.99 | 2.61 | 1.95 | 2.14 | 2.20 | 2.39 | 2.38 | 2.26 | 1.36 | 2.66 | 2.11 |

## 4. Discussion

Chlorothalonil, normally applied multiple times within one cropping season, is easily adsorbed onto and bound to the soil matrix [21,22]. Although chlorothalonil is degraded relatively quickly in soil, repeated chlorothalonil applications under greenhouse conditions have been shown to lead to its accumulation (as well as that of its metabolite) in soil, thereby altering soil microbial activities [5]. Actually, the consequences of successive applications of chlorothalonil to soil differ from the effects of a single application [4]. Soil respiration was permanently inhibited by the chlorothalonil at 25 mg·kg$^{-1}$. Dehydrogenase activity (DHA) was reduced significantly on day 15 after four successive treatments of 10 mg·kg$^{-1}$ and 25 mg·kg$^{-1}$ of chlorothalonil. Compared with that, in the present study, results of the general linear model analysis showed that different chlorothalonil treatments strongly affected soil respiration, certain soil enzymatic activities, and microbial community structure in yellow–brown loam soil (Table 4). Generally, soil respiration increased with increasing concentrations of chlorothalonil. Similarly, a strong dose–response relationship between the acetamiprid exposure concentration and the inhibition of soil respiration was

found by Yao et al. [23]. Wu et al. [4] also indicated that the effect of repeated chlorothalonil applications on soil respiration and DHA was a concentration-dependent process.

**Table 4.** Correlation analysis between different chlorothalonil applied concentrations and soil respiration, enzymatic activities, and culturable microorganisms.

|  |  | Respiration | Urease | Phosphatase | Saccharase | Bacteria | Fungi |
|---|---|---|---|---|---|---|---|
| Treatments | F | 3.496 | 4.947 | 0.281 | 8.603 | 8.920 | 6.380 |
|  | p | 0.040 | 0.012 | 0.756 | 0.001 | 0.001 | 0.004 |

Our results indicate that in soil treated with chlorothalonil, soil respiration is related to soil microbial biomass, which agrees with Anderson and Domsch [24], who indicated that microbial biomass can be estimated using soil respiration data. The effects of fungicides on soil respiration are related to the soil type and the heterotrophic microbial activities caused by pesticide residues [25]. Although the exact mechanism of how chlorothalonil application influences the soil respiration process is unclear, the increase of $CO_2$ production may be related to the changes in biomineralization processes during the substrate-induced respiration determination in the presence of the chlorothalonil residues or its metabolite [25]. Moreover, residual chlorothalonil in soil may destroy the membranes of microbes, resulting in increased levels of organic matter that, in turn, increases soil respiration. This can be supported by the inhibition of DHA in soil occurred after the application of chlorothalonil, which has been proven in a previous study [4]. DHA is usually used as an indicator of overall microbial activity because it occurs in soils as integral parts of intact cells and is linked to soil microbial oxidation of organic substances [26]. The data obtained in the present study showed that the stimulative effects on soil respiration could persist during the 60 d of observation in Ch 300 group, showing the interference of the fungicide on soil-oxidative processes [25,27,28]. Soil enzyme activities as "sensors" of soil integration information are considered, not only from soil physico-chemical status, but also based on the soil microbial status. Su et al. [28] suggest that chlorothalonil in soil may influence soil denitrification function by directly reducing microbial electron transport system activity, thus lowering levels of the electron donor nicotinamide adenine dinucleotide (NADH) and ATP. The activities of specific soil enzymes are influenced by changes in the microbial community that synthesizes these enzymes [29–31]. In the present study, we observed that soil applied with chlorothalonil significantly decreased levels of saccharase activity compared to the control; i.e., saccharase activity was negatively correlated to chlorothalonil treated concentration. In contrast, the significantly increased soil phosphatase activity and decreased soil urease activity were only observed at a high concentration of residual chlorothalonil (Ch 300) in general. These results suggest that certain enzymes are insensitive to low concentration of residual chlorothalonil in soil.

The patterns of soil microbial biomass and soil enzyme activities were similar, confirming the close relationship between both factors. In many fields of microbial ecology, PCR–DGGE has been successfully used to assess the diversity of microbial communities and to determine the community dynamics in response to environmental variations [32]. The PCR–DGGE patterns generated by the different treatments indicate the presence of a higher number of different microbial taxa in our study, which help us to infer that chlorothalonil adversely influenced microbial biomass and diversity. The soil PCR–DGGE profile shows that the general structures of the bacterial and fungal communities in soil changed in response to residual chlorothalonil. A great community change in the present study, particularly at high concentrations of chlorothalonil, probably resulted from suppressed toxicity of the immunostimulated reactive oxygen species (ROS) and baseline nicotinamide adenine dinucleotide phosphate (NADPH) concentration in soil microbe [33]. It may be possible to manage the toxicity of chlorothalonil to soil enzymes and microbial communities; for example, such toxicity may be reduced by probiotic strains such as *Stenotrophomonas acidaminiphila* BJ1 [34]. The combination of soil enzymatic activity and PCR–DGGE analyses

is believed to be a better approach to assess soil quality [35]. Therefore, we combined soil property and PCR–DGGE analyses to estimate the yellow–brown loam soil quality in the present study. Although the key role played by microorganisms in the soil health are clear, we have little information on the importance of microbial diversity on soil properties [35–38]. Based on our findings, application of chlorothalonil significantly affects soil enzymatic activities and microbial community structure. Further studies are needed to explore the precise mechanism by which chlorothalonil influences the properties of soil and its microbial community.

## 5. Conclusions

Chlorothalonil in yellow–brown loam soil strongly affects soil respiration, enzymatic activities, and microbial community structure. The highest inhibitory effects of chlorothalonil application on soil enzymatic activities and microbial community structure were observed immediately after application, with some recovery occurring at later stages. Changes in the microbial community structure were most evident at a high chlorothalonil treatment concentration. The combination of soil biochemical analysis and PCR–DGGE analyses provides a tool to effectively determine the effects of residual chlorothalonil on the soil microbial community.

**Author Contributions:** Conceptualization, J.J. and Y.Y.; data curation, J.J. and Y.Y.; investigation, J.J., Y.Y., L.W. and R.L.; methodology, S.C. and R.L.; project administration, J.J.; writing—original draft, J.J. and Y.Y.; writing—review and editing, J.J., S.C. and T.L. All authors have read and agreed to the published version of the manuscript.

**Funding:** This work was financially supported by the Fundamental Research Funds for the Central Public Welfare Research Institutes (GYZX220202) and the National Key Research and Development Program of China (2018YFC1801100).

**Institutional Review Board Statement:** Not applicable.

**Informed Consent Statement:** Not applicable.

**Data Availability Statement:** The data presented in this study are available upon request from the corresponding author.

**Acknowledgments:** The authors would like to thank Yong Jia for providing the experimental plot and for his collaboration.

**Conflicts of Interest:** The authors declare no conflict of interest.

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
