# Peer review of "Effects of Chlorothalonil Application on the Physio-Biochemical Properties and Microbial Community of a Yellow–Brown Loam Soil"

_agriculture, doi:10.3390/agriculture12050608_

Round 1

Reviewer 1 Report

The study is about the evaluation of the effect of chlorothalonil residues on a specific soil. The time of dissipation of the chlorothalonil, and the effect of the fungicide on enzyme activities, “soil respiration”, and microbial community alterations were evaluated. The importance of the study relies on the generation of new knowledge about chlorothalonil contamination on yellow-brown loam soil, showing alterations in the parameters tested when the higher chlorothalonil concentration was utilized.

The manuscript needs some corrections presented below.

Line 23 - The keywords that already appear in the title could be changed.

Lines 32 – 33 It would be more impacting if values were added for “multiple times” and “short duration”.

Line 82 – The incubation temperature is an important parameter, so I suggest the alteration of “room temperature” to the temperature value and possible temperature variations.

Line 86 – The acronym PCR-DGGE appears for the first time in the manuscript so it should be defined (the abstract section is independent).

Lines 111-116 – All descriptions of the enzymatic activities could be improved. There is information that should be presented, such as how the assays were measured, how much soil was utilized in each assay, or how controls were made.

Line 114 – The saccharase activity assay description is lacking information and the reference stated seems to be for a different method (without 3,5-dinitrosalicylic acid).

Line 121 – The period and temperature of incubation are important information.

Line 127 – alter “sofeware” to “software”.

Line 129 – The expression “The coefficient of DNA sequence similarity” sound strange, because what was evaluated was the DGGE fingerprint for each sample and not the DNA sequence. Using “The similarity coefficient” seems better.

Lines 117 – 133 – In this section the description of how the dendrograms were constructed should appear, adding which method and software were used.

Line 169 - Figure 2a the axis “x” presented the values 0; 10; 20; 30; 40, while the other presented 0; 10; 20; 30; 60. Why this difference?

In figure 2, the enzymatic activity units are lacking information, and the legend is not very helpful. For example, Fig 2b shows the unit as mg.g-1, but we have no idea of “mg” of what was measured (I believe it is NH+-N) and the time of incubation is crucial (probably for 24h). So, I recommend something like “NH+-N mg. g-1. 24h-1”. The same is applied to the other enzymatic activities. Please add the statistical information in the legend.

 Lines 180-181 – Please revise this sentence, there are inconsistencies between this and what is presented in Table 2.

Lines 190 – 202 – When the expression “significantly different” is used, please add the statistical support.

Line 203, Table 3 – In table 3 there is a description of “Total species”. How do authors correlate PCR-DGGE results with the number of total species?

Line 205, Figure 3 – Please add the description of which method was utilized for dendrograms construction.

Lines 224 – 227 – In this sentence, the addition of some references will add support to the idea presented. Although the exact mode of action of the chlorothalonil is not described, there is some information about what chlorothalonil causes in microorganisms.

Line 240, Table 4 – The Table 4 description is lacking information about the statistical analysis applied. This table is poorly explored in the manuscript.

Lines 250 – 254 – It is not evident, by the results presented, that the fungicide affects ROS and NADPH concentrations. So, some reference is needed to support this statement.

Line 325 – Reference 19 is without the publication year.

Line 333 – Reference 23 is formatted incorrectly.

In conclusion, I cannot recommend the manuscript accept without major revision.

Author Response

From Dr. Jiang Jinlin

Nanjing Institute of Environmental Sciences, Ministry of Ecology and Environment, Nanjing 210042, PR China

To Reviewers

Agriculture

Dear Reviewer #1:

Thank you very much for your E-mail on April 7 and the review of our manuscript (agriculture-1666729). The attached file is the revised manuscript “Effects of Chlorothalonil Application on the Physio-biochemical Properties and Microbial Community of a Yellow-Brown Loam Soil”. According to the comments, the detailed revisions and responses are listed below point by point:

Reviewer #1:

  1. Line 23 - The keywords that already appear in the title could be changed.

Response: Line 23: According to the reviewer’s advice, the keywords have been changed to “fungicide; dissipation; soil respiration; soil enzyme; soil microbial population”.

  1. Lines 32 – 33 It would be more impacting if values were added for “multiple times” and “short duration”

Response: According to the reviewer’s advice, we examined most of the registration data for chlorothalonil on crops based the ICAMA database from China Pesticide Information Network and changed this sentence to “It is usually applied 1−3 times to the same crop within a growing season, with application interval less than 10 days”.

  1. Lines 82 – The incubation temperature is an important parameter, so I suggest the alteration of “room temperature” to the temperature value and possible temperature variations.

Response: We have added the incubation temperature value of “(25.6 ± 1.8)oC” in the text.

  1. Lines 86 – The acronym PCR-DGGE appears for the first time in the manuscript so it should be defined (the abstract section is independent).

Response: We have corrected this mistake. The sentence of “We also evaluated the changes in the soil microbial community (culturable microbial biomass and PCR–DGGE bands) via plate counting method and PCR–DGGE analysis.” has been changed to “We also evaluated the changes in the soil microbial community represented by the culturable microbial biomass and polymerase chain reaction-denaturing gradient gel electrophoresis (PCR–DGGE) bands via plate counting method and PCR–DGGE analysis, respectively.”.

  1. Lines 111-116 – All descriptions of the enzymatic activities could be improved. There is information that should be presented, such as how the assays were measured, how much soil was utilized in each assay, or how controls were made.

Response: According to the reviewer’s advice, the descriptions of the enzymatic activities have been improved. The description of “Urease activity was determined by measuring the amount of NH+-N released from urea per gram dried soil after incubation at 37°C for 24 h [15]. Phosphatase activity was determined by measuring the amount of phenol released per gram dried soil after incubation at 37°C for 24 h with disodium phenyl phosphate as a substrate [16]. Saccharase activity was determined by measuring the amount of glucose released per gram dried soil after 24 h of incubation at 37°C with 3,5-dinitrosalicylic acid as a substrate [17].” was changed to the description as follows,

“Soil urease activity was determined spectrophotometrically using urea as a substrate [15]. 1.0 g soil was placed in a 50 ml bottle, 1 mL toluene, 10 mL 10% urea solution, and 10 mL 0.1 M Na2HPO4-Citric acid buffer (pH 6.7) were added and kept in a water bath at 37°C. After 24 h incubation, 4 mL sodium phenate and 3 mL sodium hypochlorite were added, and then the solution was diluted to 20 mL with deionization water. NH4+-N concentration was determined by the spectrophotometer at a wavelength of 578 nm. A series of standard solutions of NH4Cl covering the concentrations of 0.05–2.0 mg NH4+-N per ml was prepared for calibration. Urease activity was expressed as the amount of NH4+-N released from urea per gram dried soil after incubation at 37°C for 24 h.

Soil phosphatase activity was determined using disodium phenyl phosphate as a substrate [16]. 1.0 g of soil was placed in a 50 mL bottle, and 0.2 mL of toluene, 5 mL of 0.5 % disodium phenyl phosphate solution, 5 mL of 0.1 M phosphate buffer (pH 7.0) were added to the flask, and the suspensions were kept in a water bath at 37°C for 4 h. After incubation, suspensions of 1 mL were moved to a tube (15 mL), 2.5 mL of 0.5 % 4-aminoantipyrine, 2.5 mL of 2.5 % red potassium prussiate, and 10 mL of deionization water were added. The flask was then swirled for a few seconds to mix the contents. After 30 min, the filtrate was determined by a spectrophotometer at a wavelength of 570 nm. A series of standard solutions of phenol covering the concentrations of 0.05–5.0 mg phenol per mL was prepared for calibration. Phosphatase activity was expressed as the amount of phenol released per gram dried soil after incubation at 37°C for 24 h.

Soil saccharase activity was determined using 3,5-dinitrosalicylic acid as a substrate [17]. 2.0 g of soil sample was placed in a 50 mL bottle, mixed with 15 mL of 18% sucrose solution, 5 mL of phosphate buffer (pH 5.5), and 0.2 mL of methylbenzene, and then kept in a water bath at 37°C for 24 h. Then, 3,5-dinitrosalicylic acid solution and deionized water were added into the filtrate in the tubes, and the color intensity was determined using a spectrophotometerat a wavelength of 508 nm. Simultaneously, the control with no-matrix and no-soil was made. Soil saccharase activity was expressed as the amount of glucose released per gram dried soil after 24 h of incubation at 37°C.”

We also found the method of saccharase activity determination was referred to a incorrect reference, which has been corrected in MS.

  1. Line 114 – The saccharase activity assay description is lacking information and the reference stated seems to be for a different method (without 3,5-dinitrosalicylic acid).

Response: Thank the reviewers for pointing out the incorrect reference. We have added the correct reference as follows,

“17. Guan, S.Y., Zhang, D.S., Zhang, Z.M. Soil enzymes and their research methods. Beijing: Agriculture Press, 1986”.

  1. Line 121 – The period and temperature of incubation are important information.

Response: According to the reviewer’s advice, information on the incubation period and temperature has been added in 2.6 section.

  1. Line 127 – alter “sofeware” to “software”.

Response: The “sofeware” has been corrected to “software”.

  1. Line 129 – The expression “The coefficient of DNA sequence similarity” sound strange, because what was evaluated was the DGGE fingerprint for each sample and not the DNA sequence. Using “The similarity coefficient” seems better.

Response: According to the reviewer’s advice, the “The coefficient of DNA sequence similarity” in manuscript has been altered to “The similarity coefficient”.

  1. Lines 117 – 133 – In this section the description of how the dendrograms were constructed should appear, adding which method and software were used.

Response: According to the reviewer’s advice, the analysis method and software used for construction of dendrograms have been added in 2.6 section of manuscript.

  1. Line 169 - Figure 2a the axis “x” presented the values 0; 10; 20; 30; 40, while the other presented 0; 10; 20; 30; 60. Why this difference?

In figure 2, the enzymatic activity units are lacking information, and the legend is not very helpful. For example, Fig 2b shows the unit as mg.g-1, but we have no idea of “mg” of what was measured (I believe it is NH+-N) and the time of incubation is crucial (probably for 24h). So, I recommend something like “NH+-N mg. g-1. 24h-1”. The same is applied to the other enzymatic activities. Please add the statistical information in the legend.

Response: The incorrect “x value” has been corrected in Figure 2a: “40” changed to “60”. We have revised the all Figures according to the reviewer’ suggestion.

  1. Lines 180-181 – Please revise this sentence, there are inconsistencies between this and what is presented in Table 2.

Response: “The level of culturable fungal biomass in Ch 3 was significantly lower than that of the control and Ch 300 after 10 days (p < 0.05)” has been corrected to “The level of culturable fungal biomass in Ch 300 was significantly lower than that of the control and Ch 3 after 10 days (p < 0.05)”.

  1. Lines 190 – 202 – When the expression “significantly different” is used, please add the statistical support.

Response: According to the reviewer’s advice, we have revised the related description in the manuscript highlighted with yellow highlight boxes.

  1. Line 203, Table 3 – In table 3 there is a description of “Total species”. How do authors correlate PCR-DGGE results with the number of total species?

Response: Here the total species number represent the number of digitized DGGE bands calculated by Gel Compare II software (although each strip may contain more than one microbes species) and we have add the related description in manuscript. The Shannon index was also calculated by the Shannon program in Gel Compare II software according to the following formula,

Where s represents the number of strips per lane; pi is the proportion of the gray level (height of the peak) of the ith strip in a lane to the total gray level in this lane.

  Because the calculation method is embedded in the model software, the specific calculation is not repeated in the text for the sake of brevity.

  1. Line 205, Figure 3 – Please add the description of which method was utilized for dendrograms construction.

Response: Please refer to the response to Q10. We have added the description in 2.6 Section.

  1. Lines 224 – 227 – In this sentence, the addition of some references will add support to the idea presented. Although the exact mode of action of the chlorothalonil is not described, there is some information about what chlorothalonil causes in microorganisms.

Response: According to the reviewer’s suggestion, we revised the description in Lines 224-227 to “The effects of fungicides on soil respiration are related to the soil type and the heterotrophic microbial activities caused by pesticide residues [25]. Although the exact mechanism of how chlorothalonil application influences the soil respiration process is unclear, the increase of CO2 production may be relate to the changes in biomineralization processes during the substrate-induced respiration determination in the presence of the chlorothalonil residues or its metabolites [25]. The data obtained in the present study showed that the stimulative effects on the soil respiration could persist during the 60 d of observation in Ch 300 group, showing the interference of the fungicide on the soil-oxidative processes [25-27].”

  1. Line 240, Table 4 – The Table 4 description is lacking information about the statistical analysis applied. This table is poorly explored in the manuscript.

Response: The information about the statistical analysis applied has been addin in 2.7 section. And we improved the discussion part in the manuscript.

  1. Lines 250 – 254 – It is not evident, by the results presented, that the fungicide affects ROS and NADPH concentrations. So, some reference is needed to support this statement.

Response: Lines 250-254 has been revised and the necessary reference is added to support the statement.

  1. Line 325 – Reference 19 is without the publication year.

Response: The publication year has been updated.

  1. Line 333 – Reference 23 is formatted incorrectly.

Response: The Reference has been corrected to “Yao, X.H., Min, H., Lu, Z.H., Yuan, H.P. Influence of acetamiprid on soil enzymatic activities and respiration. Eur. J. Soil Biol. 2006, 42, 120–126”.

We have revised the whole manuscript carefully to correct grammatical and syntax errors. In addition, we asked a native English speaker who is skilled author to check the English.

All the changes in our manuscript were highlighted in yellow color. We thank the reviewer for the comments. At the same time, thank you very much for your time and assistance. I hope the manuscript has been improved satisfactorily and can be accepted for publication in Agriculture.

Sincerely yours,

Jiang Jinlin

Reviewer 2 Report

Dear Authors, I found your manuscript worthy. However, I do have some suggestions and questions. 

The manuscript claim that repeated application of Chlorothalonil can harm soil health indicators and hence are a problem. However, did you checked its results after repeated application? As far as I know from recent literature the Chlorothalonil is easily degraded in soil after application. Are your results only superficial and or circumstantial?

Moreover, have you collected the soil samples from the soils which receive multiple exposures of Chlorothalonil?

I could not find the hypothesis? What is new in your studies which makes it different from other studies in the fiekd? There should be a gap statement.

Are you sure you measured the soil respiration using IRGA? Any reference?

I suggest authors to improve the quality of the figures. Some font sizes and xy pannels. 

How many replications were there for each treatment? n-???? add this info in each table and figure caption. 

Satisfied with discussion and conclusion

Author Response

From Dr. Jiang Jinlin

Nanjing Institute of Environmental Sciences, Ministry of Ecology and Environment, Nanjing 210042, PR China

To Reviewers

Agriculture

Dear Reviewer #2:

Thank you very much for your E-mail on April 7 and the review of our manuscript (agriculture-1666729). The attached file is the revised manuscript “Effects of Chlorothalonil Application on the Physio-biochemical Properties and Microbial Community of a Yellow-Brown Loam Soil”. According to the comments, the detailed revisions and responses are listed below point by point:

Reviewer #2:

  1. The manuscript claim that repeated application of Chlorothalonil can harm soil health indicators and hence are a problem. However, did you checked its results after repeated application? As far as I know from recent literature the Chlorothalonil is easily degraded in soil after application. Are your results only superficial and or circumstantial?

Response: Thank you for the reviewer’s comment. We totally agree with you that chlorothalonil is relatively easy to degraded in soil after single application. In this study, approximately 98.8% and 96.2% of the initial chlorothalonil concentrations dissipated within 30 days in Ch 3 and Ch 300, respectively. It seemed chlorothalonil did not accumulate significantly in soil after single applications of 3.0 or 300 mg·kg−1 of dry soil. However, repeated application (2-3 times) of this fungicide is shown a more common way used in reality, sometimes it can reach 5 times the recommended dosage per growing season or year (Chaves et al., 2007; Wu et al., 2014). The repeated chlorothalonil applications lead to its accumulation (as well as that of its metabolite) in soil, thereby altering soil microbial activities (Wu et al., 2014). Actually, the consequences of successive applications of chlorothalonil to soil differ from the effects of a single application (Wu et al., 2014). We have compared our result from the single application with the results from the successive applications of chlorothalonil to soil in Discussion. And it should be noted that although the fungicide may be easily degraded in soil, its main metabolite is 30 times more toxic and was more persistent than the parental molecule (Cox, 1997). These characteristics reinforce the need of search of the accumulation of chlorothalonil and its possible impact on the physio-biochemical properties and microbial community of various soil types. The changes of the soil health indicators in this may be resulted from the direct or indirect influence of the residual chlorothalonil or its metabolite.

Reference:

  • Cox, C. Chlorothalonil. J. Pestic. Reform. 1997, 17, 14–
  • Stefani, A.Jr, Felício, J.D., de Andréa, M.M. Comparative assessment of the effect of synthetic and natural fungicides on soil respiration. Sensors (Basel). 2012, 12(3), 3243–3252.
  • Wu, X., Cheng, L., Cao, Z., Yu, Y. Accumulation of chlorothalonil successively applied to soil and its effect on microbial activity in soil. Ecotoxicol. Environ. Saf. 2012, 81, 65–69.
  • Wu, X.W., Yin, Y.M., Wang, S.Y., Wang, S.Y., Yu, Y.L. Accumulation of chlorothalonil and its metabolite, 4-hydroxychlorothalonil, in soil after repeated applications and its effects on soil microbial activities under greenhouse conditions. Environ. Sci. Pollut. Res. 2014, 21, 3452–3459.

  1. Moreover, have you collected the soil samples from the soils which receive multiple exposures of Chlorothalonil?

Response: So far little is known about the exact mechanism of residual chlorothalonil on essential soil micro-ecosystem functions and properties, including soil respiration, soil enzymatic activities, microbial biomass, and microbial community structure, both after the single application or the repeated applications. Therefore, we conducted this study is to investigate the effects of chlorothalonil after single application on the physio-biochemical properties and microbial community of a yellow-brown loam soil. We collect soil from the top layer (0–15 cm) of a botanical garden for experiment, which contained no detectable amounts of residual chlorothalonil. So we did not collected the soil samples from the soils which receive multiple exposures of chlorothalonil in the present study.

  1. I could not find the hypothesis? What is new in your studies which makes it different from other studies in the fiekd? There should be a gap statement.

Response: The characteristics of the effects of residual chlorothalonil on essential soil micro-ecosystem functions and properties, including soil respiration, soil enzymatic activities, microbial biomass, and microbial community structure, and involved mechanism are still unclear, both after the single application or the repeated applications. And there were even conflicting results from different studies. According to the reviewer’ advice, We have improve the description in Introduction.

  1. Are you sure you measured the soil respiration using IRGA? Any reference?

Response: We used a Soil Respiration Meter (China) with a dual wavelength infrared CO2 analyzer inside as shown in following picture. We confused the English name of this instrument and the incorrect name has been revised to “Soil Respiration Meter”.

  1. I suggest authors to improve the quality of the figures. Some font sizes and xy pannels. 

How many replications were there for each treatment? n-???? add this info in each table and figure caption. 

Response: According to the reviewer’ suggestion, the Figures have been improved and the information of replications for each treatment has been added in each table and figure caption.

We have revised the whole manuscript carefully to correct grammatical and syntax errors. In addition, we asked a native English speaker who is skilled author to check the English.

All the changes in our manuscript were highlighted in yellow color. We thank the reviewer for the comments. At the same time, thank you very much for your time and assistance. I hope the manuscript has been improved satisfactorily and can be accepted for publication in Agriculture.

Sincerely yours,

Jiang Jinlin

Round 2

Reviewer 1 Report

The authors made all corrections, so the manuscript could be accepted in the present form.

Reviewer 2 Report

Satisfied with the revision. Irt can be accepted.